# Volatile Short-Chain Aliphatic Aldehydes Act as Taste Modulators through the Orally Expressed Calcium-Sensing Receptor CaSR

**DOI:** 10.3390/molecules28124585

**Published:** 2023-06-06

**Authors:** Seiji Kitajima, Yutaka Maruyama, Motonaka Kuroda

**Affiliations:** Institute of Food Research & Technologies, Ajinomoto Co., Inc., 1-1 Suzuki-cho, Kawasaki-ku, Kawasaki 210-8681, Kanagawa, Japan

**Keywords:** CaSR, calcium-sensing receptor, aldehyde, isovaleraldehyde, *kokumi* substance, methional, sensory evaluation

## Abstract

Aldehydes are natural volatile aroma compounds generated by the Maillard reaction of sugars and amino acids in food and affect the flavor of food. They have been reported to exert taste-modifying effects, such as increases in taste intensity at concentrations below the odor detection threshold. The present study examined the taste-enhancing effects of short-chain aliphatic aldehydes, such as isovaleraldehyde (IVAH) and 2-methylbutyraldehyde, thus attempting to identify the taste receptors involved. The results obtained revealed that IVAH enhanced the taste intensity of taste solutions even under the condition of olfactory deprivation by a noseclip. Furthermore, IVAH activated the calcium-sensing receptor CaSR in vitro. Receptor assays on aldehyde analogues showed that C3-C6 aliphatic aldehydes and methional, a C4 sulfur aldehyde, activated CaSR. These aldehydes functioned as a positive allosteric modulator for CaSR. The relationship between the activation of CaSR and taste-modifying effects was investigated by a sensory evaluation. Taste-modifying effects were found to be dependent on the activation state of CaSR. Collectively, these results suggest that short-chain aliphatic aldehydes function as taste modulators that modify sensations by activating orally expressed CaSR. We propose that volatile aroma aldehydes may also partially contribute to the taste-modifying effect via the same molecular mechanism as *kokumi* substances.

## 1. Introduction

Flavor mainly results from the combination of three senses: taste, somatosensation, and olfaction. Taste and somatosensation are perceived in the mouth, and these are distinguished from olfaction as oral sensations. Taste, which is detected by taste receptors on the tongue, generates sensations of sweetness, sourness, bitterness, saltiness, umami, and other potential sensations [1]. Sugars and sweeteners have been shown to elicit sweetness by activating T1R2/T1R3 [2]. Furthermore, volatile flavor compounds with an odor were found to affect the taste of food products [3,4]. Previous studies demonstrated that strawberry flavor enhanced sweetness, while flavor compounds containing soy sauce enhanced saltiness [5,6,7]. In addition, volatile aroma compounds affected taste intensity at concentrations below the odor detection threshold [3,4]. Short-chain aliphatic aldehydes, such as isovaleraldehyde (IVAH) and 2-methylbutyraldehyde, are natural volatile aroma compounds, which are key flavors generated by the Maillard reaction of sugars and amino acids in the process of manufacturing food products, such as cheese, soy sauce, and sake [8,9]. The addition of these aldehydes to a basic taste solution or foods at concentrations below the odor detection threshold was found to enhance the intensities of sweet, umami, and salty tastes without imparting a smell [10,11]. A recent study reported that methional, a well known flavor compound in cheese and soy sauce with an aldehyde structure containing sulfur, functioned as a positive allosteric modulator (PAM) for the human umami receptors T1R1/T1R3 [12]. Other studies demonstrated that methional enhanced salty [13] and umami [14] taste in human sensory evaluation tests. Therefore, aldehydes, including methional, may enhance the intensities of multiple basic tastes by activating the taste receptors expressed on the tongue. However, the receptors involved in the enhancement of multiple tastes have not yet been clarified.

γ-Glutamyl peptides, including γ-glutamyl-valyl-glycine (γ-EVG) and glutathione (GSH) [15,16,17,18,19], are taste modulators that enhance the taste intensities of sweetness, saltiness, and umami, as well as the mouthfeel, such as fattiness [20] and pungency [21]. They have no taste at the concentrations used and are now known as *kokumi* substances. The addition of *kokumi* substances to basic taste solutions or foods enhances the intensities of thickness, mouthfulness, and continuity, which, in turn, increase taste intensity [17,22]. Ohsu et al. [17] and Maruyama et al. [23] suggested that the activation of the calcium-sensing receptor CaSR by *kokumi* substances is involved in the taste-enhancing effects of *kokumi* substances. CaSR is a seven-transmembrane-spanning, G protein-coupled receptor (GPCR) belonging to the same class C subfamily of taste receptors, such as T1R1 and T1R3 [24]. It is reported that CaSR is expressed in several cells and tissues, including the parathyroid gland and kidney. CaSR is also known to be involved in the regulation of calcium concentrations of the plasma and plays a central role in calcium homeostasis [25].

We hypothesized that CaSR expressed on the tongue is involved in the taste-enhancing effects of aliphatic aldehydes. Therefore, the present study investigated the relationship between the activation of CaSR and the taste-enhancing effects of aliphatic aldehydes using a sensory evaluation and cell-based CaSR receptor assay. The results obtained showed the activation of CaSR by short-chain aliphatic aldehydes and suggest that these aldehydes function as taste modulators that modify oral sensations by activating CaSR expressed in the oral cavity.

## 2. Results

### 2.1. Taste-Enhancing Effects of IVAH on Basic Taste Solution

2-AFC tests were conducted to investigate whether the addition of aldehydes to the basic taste solution enhanced taste intensity. As basic taste solutions, we used the umami and salty taste solution mixed with NaCl and MSG, used in the sensory evaluation of *kokumi* substances [17,22]. IVAH was used as the aliphatic aldehyde and was added at a concentration (2 ppb) close to the detection threshold reported by Chen et al. [9]. As shown in Figure 1A, under conditions without a nose clip, the addition of IVAH to taste solutions significantly enhanced their taste intensities. Panelists also commented that the addition of IVAH enhanced the thickness, continuity, and mouthfulness of umami and salty taste solutions, as previously reported with the addition of *kokumi* substances [17,22]. Each of these three oral sensations is defined in the papers on *kokumi* substances as follows. Thickness was expressed in terms of increased taste intensity at ~5 s after tasting; continuity was expressed as the taste intensity at ~20 s; and mouthfulness was expressed as the reinforcement of the taste sensation throughout the mouth and not just on the tongue [17]. In this condition, all panels did not mention the odor of the sample, but it is generally known that IVAH has a malty, fruity, and cocoa-like odor [26]. We investigated whether the taste-enhancing effects of IVAH were due to a cross-modal effect by olfaction using nose clips. Since it was difficult to swallow samples while wearing nose clips, evaluations were conducted using the spitting out method. As shown in Figure 1B, the taste-enhancing effects of IVAH were detected even with nose clips. These results indicate that IVAH, a volatile aliphatic aldehyde, enhanced the taste intensity of a basic taste solution by stimulating the oral cavity.

### 2.2. Human CaSR Responds to Aldehydes

A previous study reported that calcium salts, amino acids, and *kokumi* substances, such as γ-glutamyl peptides, including γ-EVG, activate human CaSR [19]. These *kokumi* substances have also been shown to enhance taste intensities when added to basic taste solutions [16,19]. To investigate whether the taste-enhancing effects of IVAH shown in Figure 1 were mediated by CaSR on the tongue, its activation by IVAH was examined using a calcium assay on human CaSR-expressing cultured cells. The results obtained showed that IVAH also activated CaSR-expressing cells, similarly to CaCl_2_ and γ-EVG, which are ligands that activate CaSR, and they increased intracellular calcium concentrations. In addition, the chelation of divalent metal ions, the agonists for CaSR, by EDTA (1 mM) or the co-addition of NPS-2143 (20 μM), a selective antagonist of CaSR, strongly attenuated the activation effect of IVAH on CaSR (Figure 2D). These results indicate that IVAH, similar to other CaSR activators, activated CaSR.

We investigated whether aliphatic aldehydes with similar structures other than IVAH activated CaSR. Aliphatic C3-C8 aldehydes with three to eight carbons were examined using the calcium assay. The results obtained showed that aliphatic C3-C6 aldehydes activated cells expressing CaSR (Figure 3A–H). Among the nine aldehydes tested, IVAH, a C5 aldehyde, was the most potent activator, activating CaSR from a low concentration (Figure 3I). We also examined the effects of other analogues, such as fatty acids or alcohols with structures similar to C3-C6 aldehydes, on CaSR. Neither fatty acids (Figure 3J), nor alcohols (Figure 3K), activated CaSR. These results suggest that C3-C6 aldehyde structures activate CaSR in a structure-dependent manner.

### 2.3. Human CaSR Responds to the Sulfur-Containing Aldehyde Methional

Methional is a C4 sulfur-containing aromatic aldehyde derived from methionine [27]. Toda et al. [12] reported that this methional acts on the human umami taste receptor T1R1 and functions as a PAM. Therefore, we investigated whether methional, which has a similar structure to aliphatic aldehydes, activated CaSR. Fatty acid (3-(methylthiol) propanic acid) and alcohol (methionol) analogues with similar structures to methional were also examined using the CaSR assay, and their functionalities were compared. The results obtained showed that only methional activated CaSR, while neither of the methional analogues tested activated CaSR (Figure 4). These results suggest that the aliphatic aldehyde structure is responsible for CaSR activation.

### 2.4. IVAH and Methional Enhanced the Responses of CaSR to Calcium Ions

The orthosteric agonists for CaSR are calcium ions. Therefore, we investigated the effects of IVAH and methional on the responses of CaSR to calcium chloride in CaSR-expressing cells. We also used γ-EVG, an activator of CaSR, and isovaleric acid (IVAA), an acid analogue of IVAH that does not activate CaSR, for comparison. The results obtained showed that IVAH, methional, and γ-EVG enhanced the response of CaSR to calcium chloride at low concentrations, indicating that they may function as a PAM for CaSR (Figure 5). In contrast, response changes were not observed with IVAA.

### 2.5. Relationship between Taste-Enhancing Effects and CaSR Activity

We investigated whether the taste-enhancing effects of IVAH were mediated by the activation of CaSR on the tongue using NPS-2143, a selective antagonist of CaSR. NPS-2143 was added to the basic taste solution, and changes in the taste-enhancing effects of IVAH were examined by 2-AFC in a sensory evaluation with nose clips. The present results showed that the addition of NPS-2143 eliminated the significant taste-enhancing effects of IVAH shown in Figure 1 (Figure 6A). We also compared the taste-enhancing effects of IVAH, which activates CaSR, as well as IVAA, which does not activate CaSR, by a sensory evaluation with nose clips. The results obtained showed that the taste solution to which IVAH, which activates CaSR, was added, had significantly stronger taste intensity than the sample to which IVAA was added (Figure 6B). These results suggest that the taste-enhancing effects of IVAH were mediated by the activation of CaSR on the tongue.

## 3. Discussion

The present study was conducted to elucidate the molecular mechanisms by which aliphatic short-chain aldehydes exert taste-enhancing effects. In sensory evaluations, IVAH was used as a typical short-chain aldehyde to confirm its taste-enhancing effects. We examined whether the addition of IVAH to the basic taste solution mixed with NaCl and MSG, which is mainly used in the sensory evaluation of *kokumi* substances, enhances the taste intensity. Although the possibility of some bias in the results of sensory study cannot be completely denied, since all of the sensory evaluations were performed by Ajinomoto employees in Japan and by trained panels engaged in taste research, the results showed that the taste intensity of the basic taste solution were significantly enhanced with and without nose clips (Figure 1). Therefore, the effects of the volatile aroma compound IVAH on the oral cavity appear to contribute to its taste-enhancing effects. IVAH in the sensory evaluations was added at a concentration close to the odor perception threshold reported by Chen [9] and is tasteless. Therefore, the functionality of IVAH appears to be similar to that of *kokumi* substances, which have no taste by themselves, but enhance taste intensity when added to a basic taste solution, such as umami and salty taste solution, as reported by Ueda et al. [17] and Ohsu et al. [17]. On the other hand, this result does not exclude that the odor of IVAH enhanced taste but suggests that part of its taste-enhancing effect is elicited by oral stimuli, as shown in Figure 1B, in which the percentage of panelists selecting the IVAH sample decreased when a nose clip was attached.

GSH is a tripeptide that has been identified as a *kokumi* substance. Similar to other γ-glutamyl peptides, GSH has been shown to enhance the intensities of the basic tastes by activating CaSR [19], which is expressed on taste cells in the tongue [23]. Therefore, we investigated whether IVAH and other short-chain aliphatic aldehydes activated CaSR, as well as GSH and other γ-glutamyl peptides, using a cell-based assay on CaSR-expressing cells. As shown in Figure 3, C3-C6 aldehydes selectively activated CaSR. The selective inhibitor of CaSR, NPS-2143, inhibited its activation, suggesting the carbon number-dependent activation of CaSR by aliphatic aldehydes (Figure 2). Although CaSR was previously shown to be activated by a wide spectrum of amino acids, there is no evidence to support its activation by aldehydes. IVAH is generated from leucin in fermented food. On the other hand, leucin and aliphatic amino acids have not been reported to activate CaSR [28]. CaSR may be involved in the sensing of not only amino acids, but also their metabolites, and it may also be indirectly involved in the sensing of these amino acids. Although we focused on aliphatic aldehydes in the present study, CaSR is also activated by aromatic amino acids, such as tyrosine and phenylalanine [29]. Therefore, future studies on the activation function of CaSR by aromatic aldehydes will provide a more detailed understanding of the relationship between chemical structures and their activation.

As shown in Figure 5, IVAH, as well as the γ-Glu peptide, γ-EVG, exhibited activity as a PAM that enhanced the responsiveness of CaSR to calcium. In addition, the sulfur-containing volatile compound, methional, which has a similar structure to aliphatic short-chain aldehydes that activate CaSR, activated CaSR and functioned as a PAM of CaSR. It is reported that there are several types of PAM, such as affinity modulation and efficacy modulation types [30]. In this study, we observed co-addition of IVAH or methional activated the CaSR responses for calcium only at low concentrations without altering the maximal response intensity. Therefore, IVAH and methional may be a type of modulator that enhance the affinity of calcium for CaSR by conformational changes, influencing the orthosteric binding pocket, which has already been suggested in aromatic amino acids [29,31]. The activation of CaSR by amino acids requires binding to the extracellular region near the calcium-binding domain [32]. Toda et al. reported that methional bound to the transmembrane region of human T1R1, a constituent molecule of the human umami taste receptor, functioned as a PAM [12]. Class C GPCRs are mainly activated by amino acids and their analogues. They are characterized by an extracellular region called the Venus Fly Trap module. The amino acid sequence of the extracellular region is diverse among receptor gene species and is considered to significantly contribute to ligand selectivity [33]. On the other hand, the transmembrane region was found to have higher homology than the extracellular region [12,34]. Therefore, short-chain aliphatic aldehydes, including methional, may also bind to the transmembrane membrane region of CaSR and function as PAMs. Therefore, further studies are needed to elucidate the binding site and mode of action responsible for the activation of CaSR.

The relationship between the activation of CaSR of aldehydes observed in vitro and taste-enhancing effects in the sensory evaluation was investigated in Figure 6A. The co-addition of the CaSR antagonist NPS-2143 abolished the taste-enhancing effects of IVAH. Furthermore, IVAH exerted significantly stronger taste-enhancing effects than IVAA, which does not activate CaSR. These results suggest that the activation state of CaSR correlates with taste-enhancing effects in human sensory evaluations, indicating that the activation of CaSR expressed in the oral cavity is involved in the taste-enhancing effects of IVAH. The concentration of IVAH at which taste-enhancing effects was observed in the sensory evaluation in the present study was very low at 2 ppb (0.02 μM), which is different from the concentration that activates CaSR in vitro. Since IVAH functions as a PAM for CaSR, its ability to activate CaSR depends on the concentration of calcium. Furthermore, there are other amino acids in saliva that activate CaSR, and these amino acids may act synergistically and affect the functionality of IVAH.

CaSR is also activated by aromatic amino acids, such as tyrosine and phenylalanine. Therefore, further studies on the activation of CaSR by aromatic aldehydes will provide more detailed information on the relationship between chemical structures and their activation. Besides amino acids, a number of components in food are known to activate CaSR, including metals, such as magnesium, poly-cations (spermine) [35], peptides derived from soybeans (β51-63) [36,37], and pH [38]. Furthermore, in the present study, CaSR was activated by volatile aldehydes; therefore, a number of food ligands may act on CaSR. Further research is needed to establish whether there are differences in the ligand-binding sites of these components and the resulting taste characteristics. In addition, it is reported that IVAH, which activated CaSR in this study, is also detected in plasma [39]. Therefore, it is suggested that IVH may be involved in calcium homeostasis via CaSR, which plays an important role in the regulation of calcium concentrations in plasma. The physiological benefits of the sensing of these various ligands by CaSR expressed on taste cells and other tissues also warranted further study.

Aldehydes are generated during the fermentation process of fermented foods, such as cheese and sake [8]. These components may enhance the taste of food and contribute to their palatability through their function as *kokumi* substances via the activation of CaSR in the oral cavity, in addition to their odor. *Kokumi* substances modify mouthfeel sensations, such as thickness, continuity, and mouthfulness [17], and the functions of these aldehydes in food as *kokumi* substances may play a role in taste sensations, such as richness perceived in fermented foods. *Kokumi* substances have been reported to enhance fat-like oreosensation [20,40]. Further studies are needed to investigate the effects on other tastes and mouthfeel sensations.

## 4. Materials and Methods

### 4.1. Chemicals

Commercially available food additive products were used in the human sensory test. Food-grade NaCl was purchased from Naikaisyoji (Tokyo, Japan), and monosodium glutamate (MSG) was purchased from Ajinomoto Co., Inc. (Tokyo, Japan). All flavor compounds (of food additive grade), including propionaldehyde, butanal, isobutylaldehyde, 2-methylbutylaldehyde, pentanal, IVA, hexanal, heptanal, octanal, propionic acid, 2-methylbutyric acid, isovaleric acid, hexanoic acid, heptanoic acid, propyl alcohol, 2-methyl-1-butanol, hexanol, heptanol, methional, methionol, and 3-(methiylthiol) propionic acid, were purchased from Sigma-Aldrich (St. Louis. MO, USA). Deionized water for test solutions and mouth rinsing was supplied by a Mill-Q water purification system (Millipore, Bedford, MA, USA). γ-Glu-Val-Gly was obtained from Ajinomoto Co., Inc. (Tokyo, Japan). NPS-2143, a CaSR inhibitor, was synthesized, as described by Rybczynska et al. [41]. Chemicals for cell based-assays, such as CaCl_2_, ethylenediaminetetraacetic acid (EDTA), and probenecid, were purchased from Sigma-Aldrich (St. Louis. MO, USA).

### 4.2. Measurement of CaSR Activity in CaSR-Expressing PEAK^rapid^ Cells

CaSR activity was assessed using PEAK^rapid^ cells by the method of Ohsu et al. [19]. Briefly, human CaSR cDNA was inserted into pcDNA 3.1 expression vectors in Opti-MEM I medium (Thermo Fisher Scientific, Carlsbad, CA, USA), mixed with FuGENE 6 (Roche Applied Science, Penzberg, Germany), and poured onto PEAK^rapid^ cells grown at a submaximum concentration. After a 24-h culture in 96-well plates, the cells were incubated with 5 μM Calcium-5 (Calcium-5 assay kit, Molecular Devices, San Jose, CA, USA) for 45–60 min, and measurements were performed using FDSS/µCELL (Hamamatsu Photonics, Hamamatsu, Japan) and its associated software. The activation of CaSR expressed in cells increases intracellular calcium ions, which is measured using Calcium-5 dye. The binding of Ca^2+^ ions to Calcium-5 increases dye fluorescence, excited at 485 nm and emitted at 525 nm. Test compounds were dissolved and administered in assay buffer (in mM): 146 NaCl, 1 MgSO_4_, 20 HEPES, 1 CaCl_2_, 2.5 Probenecid, and 1.39 glucose, pH 7.4. Transiently transfected cells were challenged with compounds and compared to vehicle alone and/or mock-transfected cells. The concentration dependence of fluorescence intensity (influx of Ca^2+^) was analyzed using FDSS/µCELL at 1-s intervals for 180 s. Regarding dose–response curve calculations, peak fluorescence responses after the addition of compounds were corrected for, normalized to background fluorescence (ΔF/F = (F − F0)/F0), and baseline noise was subtracted. Dose–response curves were calculated by a non-linear regression using GraphPad PRISM 8.0.

### 4.3. Sensory Evaluation

The protocol for the human sensory evaluation was approved by the Ethics Board of the Institute of Food Sciences & Technologies and the Institute for Innovation, Ajinomoto Co., Inc., Tokyo, Japan, and it was conducted according to the ethical guidelines of the 2017 Helsinki Declaration. All sensory evaluations were performed between AM 11:00 and AM 12:00 or between PM 3:00 and PM 4:00 in a partitioned booth at 25 °C in an air-conditioned sensory evaluation room. All panelists were employees of Ajinomoto Co., Inc., Tokyo, Japan. Panelists have significant experience in evaluating the taste intensities of various food products.

For the alternative forced-choice test (2-AFC), a basic taste solution containing a mixture of NaCl (0.5%) and MSG (0.1%) was used. A pair of 20 mL test solutions, one with and one without the test compounds, were prepared, labeled with three random numbers, and presented to the panelists at room temperature. Sixteen panelists were asked to rate the perceived taste intensity, including umami and salty tastes, and then they selected the sample with the most intense umami and salty tastes. Except for the experiment shown in Figure 1A, to avoid the effects of cross-modal interactions with odorants, panelists used nose clips. In evaluations with nose clips and those of samples containing NPS-2143, the solution was placed into the mouth and then spat out. A binomial analysis was used to establish whether a significant number of panelists identified the sample with the test compound as that with strongest taste intensity. A significance level of *p* = 0.05 was set, following the binomial statistical analysis.

## 5. Conclusions

We examined the taste-enhancing effects of volatile aroma short-chain aliphatic aldehydes and attempted to identify the taste receptors involved in these effects. The present results showed that IVAH enhanced the taste intensities of salty and umami taste solutions, even under the condition of olfactory deprivation by nose clips. We also confirmed that C3-C6 aldehydes, containing IVAH, activated the calcium-sensing receptor CaSR in vitro. Moreover, these aldehydes, containing a sulfur aldehyde, functioned as PAMs for CaSR. We showed that taste-enhancing effects were dependent on the activation state of CaSR. Collectively, these results suggest that short-chain aliphatic aldehydes function as taste modulators of oral sensations by activating CaSR expressed in the oral cavity and also propose that volatile aroma aldehydes in foods may also partially contribute to the taste-enhancing effect via the same molecular mechanism as *kokumi* substances.

## Figures and Tables

**Figure 1 molecules-28-04585-f001:**
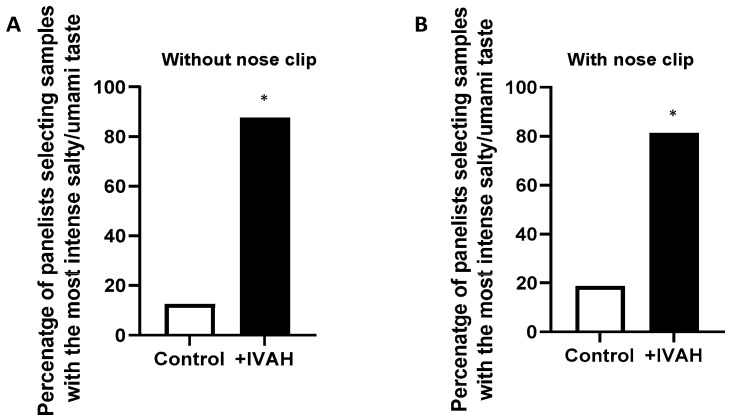
Taste-enhancing effects of isovaleraldehyde (IVAH) on basic taste solution. (**A**) The taste-enhancing effects of IVAH on salty and umami taste solution mixed with NaCl (0.5%) and MSG (0.1%) were assessed by the two-alternative forced-choice test (2-AFC). Sixteen panelists without the nose clip were presented with a taste solution alone or that plus 2 ppb IVAH in three random numbers and were asked to select the solution with the most intense salty and umami tastes. Data are shown as the percentage of panelists (out of 16) that selected a given solution as that with the most intense taste. We found that 87.5% of panelists selected the solution with IVAH as that with the most intense taste (*p* < 0.05). (**B**) The taste-enhancing effects of IVAH on salty and umami taste solutions with nose clips. Panelists with nose clips were presented with the same taste solution alone or that plus IVAH and were asked to select the solution with the most intense salty and umami tastes. Data are shown as the percentage of panelists (out of 16) that selected a given solution as that with the most intense taste. We found that 81.3% of panelists selected the solution with IVAH as that with the most intense taste (*p* < 0.05). The significance of differences is indicated above the bar (* *p* <0.05) and was evaluated by a binomial analysis.

**Figure 2 molecules-28-04585-f002:**
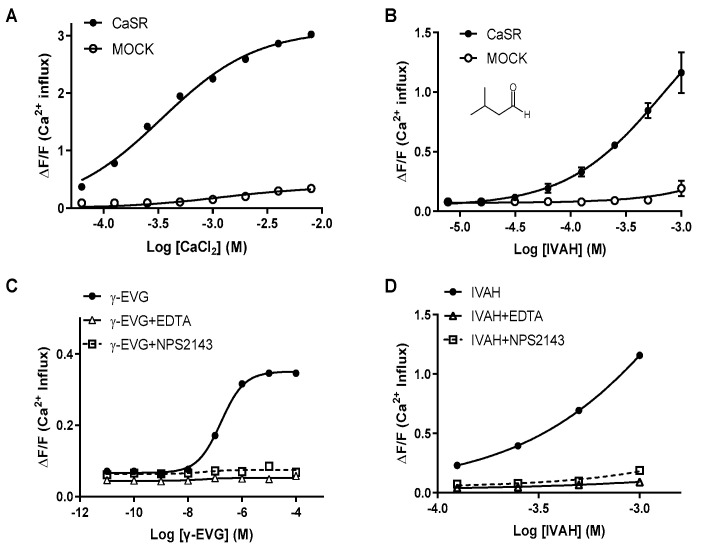
Human CaSR responds to IVAH. Dose–response curves of CaSR stimulated with CaSR ligands and IVAH. PEAK^rapid^ cells transiently transfected with human CaSR cDNA (black circles, black line) or MOCK (open circles, black line = negative control) were stimulated with increasing concentrations of each compound. Changes in fluorescence (*y*-axis, ΔF/F) were plotted against agonist concentrations (*x*-axis, logarithmically scaled). (**A**) CaSR stimulated with calcium chloride resulted in receptor activation (EC_50_ = 338 μM); (**B**) CaSR stimulated with IVAH resulted in receptor activation (EC_50_ = 681 μM); (**C**) Dose-response curves for γ-EVG were prepared, and the effects of 1 mM EDTA and 20 μM NPS2143 were examined. CaSR stimulated with γ-EVG resulted in receptor activation (EC_50_ = 0.167 μM). CaSR activation by γ-EVG was attenuated by the co-addition of 1 mM EDTA (open triangles, black line) or 20 μM NPS2143 (open squares, black dashed line); (**D**) Dose–response curves for IVAH were prepared, and the effects of EDTA (open triangles, black line) and NPS2143 (open squares, black gray dashed line) were examined. CaSR activation by IVAH was attenuated by the co-addition of EDTA or NPS2143. Data are presented as the mean ± standard error.

**Figure 3 molecules-28-04585-f003:**
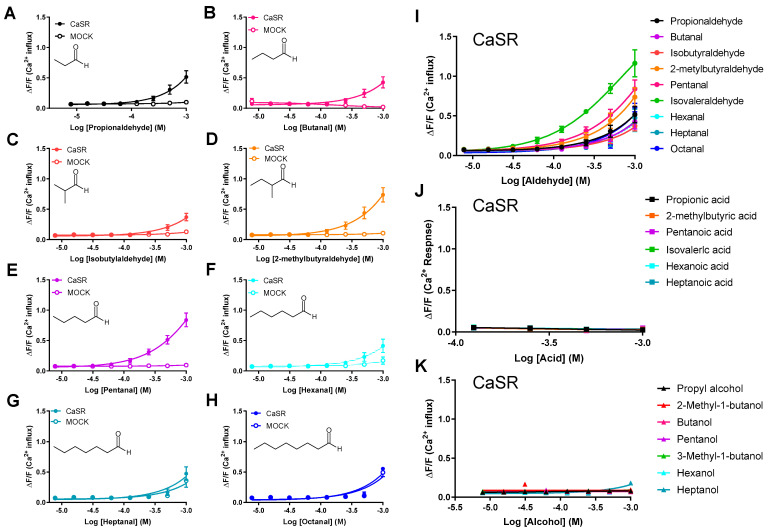
Human CaSR responds to aldehydes. Dose–response curves of CaSR stimulated with C3-C8 aldehydes, acid analogues, and alcohol analogues. PEAK^rapid^ cells transiently transfected with human CaSR cDNA (filled circles) or MOCK (open circles) were stimulated with increasing concentrations of each aldehyde, acid analogue (filled triangles), and alcohol analogue (filled squares). Changes in fluorescence (*y*-axis, ΔF/F) were plotted against agonist concentrations (*x*-axis, logarithmically scaled). CaSR was activated by propionaldehyde (**A**,**I**), butanal (**B**,**I**), isobutylaldehyde (**C**,**I**), 2-methylbutylaldehyde (**D**,**I**), pentanal (**E**,**I**), hexanal (**F**,**I**), heptanal (**G**,**I**), octanal (**H**,**I**), and IVAH (**I**). CaSR stimulated with heptanal (**G**), octanal, (**H**), acid analogues (**J**), and alcohol analogues (**K**) did not result in receptor activation. Data are presented as the mean ± standard error.

**Figure 4 molecules-28-04585-f004:**
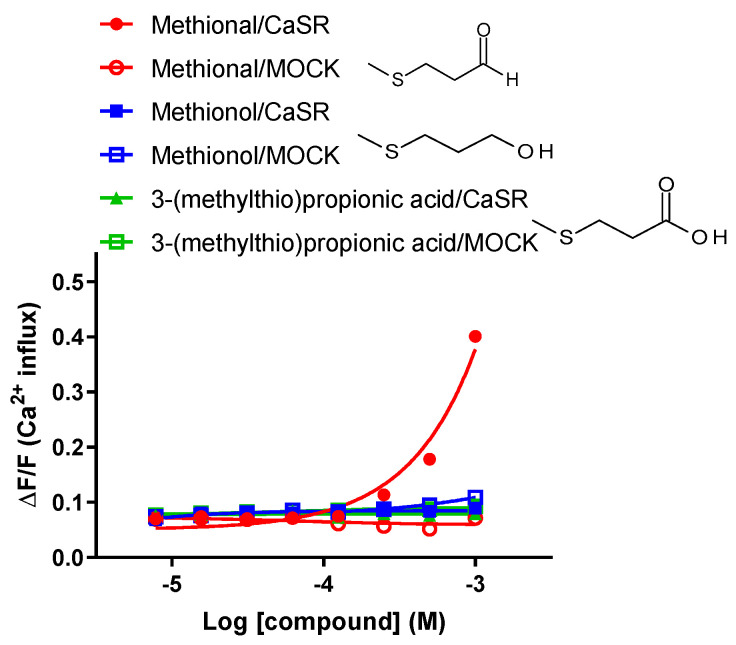
Human CaSR responds to the sulfur-containing aldehyde methional. Dose–response curves of CaSR were stimulated with methional, its alcohol analogue methionol, and the acid analogue 3-(methylthiol)propanic acid. PEAK^rapid^ cells transiently transfected with human CaSR cDNA (filled marks) or MOCK (open marks) were stimulated with increasing concentrations of methional, methionol (filled squares), and 3-(methylthiol)propanic acid (filled triangles). Changes in fluorescence (*y*-axis, ΔF/F) were plotted against agonist concentrations (*x*-axis, logarithmically scaled). CaSR was activated by methional. Methionol and 3-(methylthiol)propanic acid did not result in receptor activation. Data are presented as the mean ± standard error.

**Figure 5 molecules-28-04585-f005:**
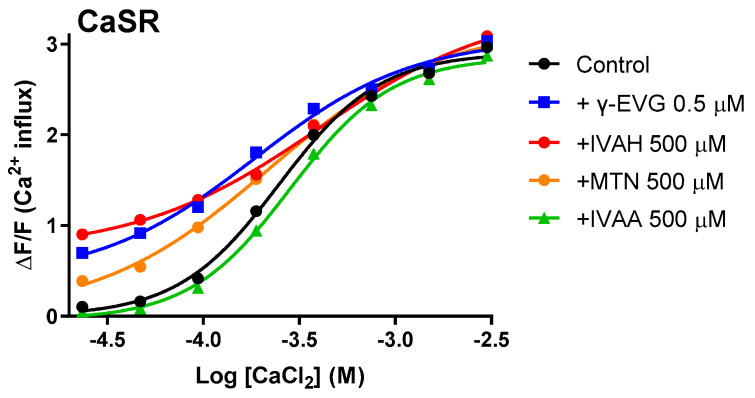
Aldehydes act as positive allosteric modulators for human CaSR. Dose–response curves of CaSR to calcium chloride were obtained in the presence and absence of IVAH (500 μM), methional (MTN, 500 μM), γ-EVG (0.5 μM), and isovaleric acid (IVAA, 500 μM). PEAK^rapid^ cells transiently transfected with human CaSR cDNA were stimulated with increasing concentrations of calcium chloride in the presence of DMSO (black circles, negative control), IVAH (red circles), methional (MTN, 500 μM, orange circles), γ-EVG (0.5 μM, blue squares), and isovaleric acid (IVAA, 500 μM, green triangles). Changes in fluorescence (*y*-axis, ΔF/F) were plotted against agonist concentrations (*x*-axis, logarithmically scaled). IVAH, methional, and γ-EVG enhanced the response of CaSR to calcium chloride, whereas isovaleric acid did not.

**Figure 6 molecules-28-04585-f006:**
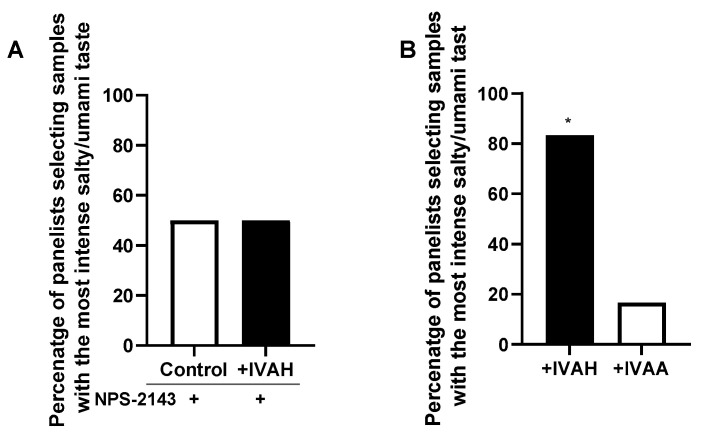
(**A**) Relationship between taste-enhancing effects and CaSR activity. The taste-enhancing effects of IVAH on the basic taste solution in the presence or absence of NPS-2143 (5 ppm, 10.4 μM) were assessed by 2-AFC. Sixteen panelists with nose clips were presented with the basic taste solution alone or that plus 2 ppb IVAH with three random numbers and were asked to select the solution with the most intense salty and umami tastes. Data are shown as the percentage of panelists (out of 16) that selected a given solution as that with the most intense taste. No significant differences were observed in the perceived intensities of the salty and umami tastes. (**B**) The taste-enhancing effects of IVAH and IVAA were compared. Compounds with higher taste-enhancing effects on salty and umami taste solutions were examined by 2-AFC. Twelve panelists with the nose clip were presented with the basic taste solutions with 2 ppb IVAH or IVAH in three random numbers and were asked to select the solution with the most intense salty and umami tastes. Data are shown as the percentage of panelists (out of 12) that selected a given solution as that with the most intense taste. We found that 83.3% of panelists selected IVAH-added solution as that with the most intense taste (*p* < 0.05). The significance of differences is indicated above the bar (* *p* < 0.05) and was evaluated by a binomial analysis.

## Data Availability

Data are contained within the article.

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
