# Peer review of "Volatile Short-Chain Aliphatic Aldehydes Act as Taste Modulators through the Orally Expressed Calcium-Sensing Receptor CaSR"

_molecules, 2023, doi:10.3390/molecules28124585_

Round 1

Reviewer 1 Report

A very comprehensive study on the effects of Strecker aldehydes on taste quality based on sensory evidence and mechanistic receptor experiments. In particular, the experiments using inhibitors of CaSR in sensory studies show clear additional evidence that CaSR activation plays a role in taste sensation.

In the introduction and the discussion, the role of CaSR on other tissues and for inflammation should be considered/discussed. If the aldehydes are real agonists/PAMs they may have also an impact on airway diseases where "calcilytic" CaSR antagonists are used as therapeutics.

Fig 1.: Did you consider statistical analysis/ANOVA between both treatments (w/o noseclip)?

Fig 2: the grey lines and dots are very difficult to read; please use colors instead.

Fig. 3:  the same for yellow lines in B, E...

Fig. 5: please explain in more detail your conclusion that the set of dose response curves are showing an allosteric effect. I am wondering why the base line Ca-response is increasing and the total intensity is remaining. For readers unfamiliar with the exact details of the operational model of allosteric modulation it will be difficult to understand. You can also reference to 10.1038/s41598-020-71228-y or an other adequate paper.

Did you also consider that the aldehydes may act as "suicide agonists", i.e. binding via e.g. lysine to the (allosteric, orthosteric?) active site changing the activation pattern? Do the receptors really recover after experiments? This suicide agonist phenomen is known from isothiocyanate activation of TRPA1 channels.

Reviewer 2 Report

The MS is a very important and timely required work. The study is clearly written and well-planned.

These investigators conducted studies in order to shed light on whether short-chain aliphatic aldehydes exert the taste-enhancing effects. They reported that one of the two aldehydes tested, i.e., isovaleraldehyde (IVAH) activated the calcium-sensing receptor CaSR in vitro.   The relationship between the activation of CaSR and taste-modifying effects was investigated by a sensory evaluation (2-AFC). They conducted studies in vitro on the cells expressing CaSR. They concluded that volatile aldehydes functioned as positive allosteric modulator of CaSR.

The title of the MS should be changed as the investigators only assessed two taste qualities (umami and salty). It should be as follows : Volatile short-chain aliphatic aldehydes act as umami and salt taste modulators through the orally expressed calcium-sensing receptor CaSR.

It should be explained why the authors did not assess other taste qualities like bitter and fat as the CaSR can be a universal taste regulator since lysine can affect fat taste perception (see Barbarossa-Thomassini et al., PlosOne).

no comments.

Reviewer 3 Report

The authors of this manuscript report interesting and novel findings where some volatile compounds (IVAH, C3-C6 and methional) activate CaSR in the oral cavity and enhance salty and umami perceptions in the same manner as kokumi substances. The findings are clear and the conclusions match the scope of the study. 

There are a few issues with the description of the study methods which need to be clarified. Specific comments below.

Major comments

Lines 112-114: Panellists being employees of Ajinomoto may be a source of bias in the study. Employees may know the purpose of the study or the substances being tasted. Secondly, panellists having experience with evaluating taste intensities may not be a true representation of the wider population. This should be addressed as a limitation of the study. Furthermore, please state the number of participants.

Lines 115-117: Please clarify the sensory evaluation method used. “Three random orders” is not clear. Also, make it clearer that NaCl and MSG were used in two separate solutions. 

Lines 117-118: Make it clear that to assess cross-modal interactions with odorants, panellists performed the test with and without nose clips. As it is written, it reads that the panellists only used nose clips. 

Lines 135-136: The odour of IVAH is noted by the authors. Were there any test measures to determine whether participants could detect these odours, with or without the nose clips?

Lines 136-140: It is unclear that participants did not wear nose clips for certain tests. Please clarify.

Minor comments

Lines 110-111: Please confirm if 12am or 12pm.

Line 134: Define the sensations, particularly continuity and mouthfulness. 

Line 204: Please check the description for figure I, as it includes more than just IVAH.  

Some grammatical errors throughout. 

Some grammatical errors throughout. Will require some minor proofreading and revisions. 
